# Role of physiotherapy in supporting recovery from breast cancer treatment: a qualitative study embedded within the UK PROSPER trial

Sophie Rees ,[1] Bruno Mazuquin ,[1,2] Helen Richmond,[1,3]
Esther Williamson ,[4] Julie Bruce ,[1] On behalf of UK PROSPER Study Group

¹Warwick Clinical Trials Unit, University of Warwick Medical School, Coventry, UK
²Faculty of Health, Psychology and Social Care, Manchester Metropolitan University, Manchester, UK
³Faculty of Medicine, Memorial University of Newfoundland, St John's, Newfoundland, Canada
⁴Nuffield Department of Orthopaedics, Rheumatology and Musculoskeletal Sciences, The Centre for Rehabilitation Research, University of Oxford, Oxford, UK

**Correspondence to**
Dr Sophie Rees;
S.Rees.1@warwick.ac.uk

## ABSTRACT

**Objectives** To explore the experiences of women with breast cancer taking part in an early physiotherapy-led exercise intervention compared with the experiences of those receiving usual care. To understand physiotherapists' experience of delivering the trial intervention. To explore acceptability of the intervention and issues related to the implementation of the Prevention Of Shoulder Problems (PROSPER) programme from participant and physiotherapist perspective.

**Design** Qualitative semistructured interviews with thematic analysis.

**Setting** UK National Health Service.

**Participants** Twenty participants at high risk of shoulder problems after breast cancer surgery recruited to the UK PROSPER trial (10 each from the intervention arm and control arm), and 11 physiotherapists who delivered the intervention. Trial participants were sampled using convenience sampling. Physiotherapists were purposively sampled from high and low recruiting sites.

**Results** Participants described that the PROSPER exercise intervention helped them feel confident in what their body could do and helped them regain a sense of control in the context of cancer treatment, which was largely disempowering. Control arm participants expressed less of a sense of control over their well-being. Physiotherapists found the exercise intervention enjoyable to deliver and felt it was valuable to their patients. The extra time allocated for appointments during intervention delivery made physiotherapists feel they were providing optimal care, being the 'perfect physio'. Lessons were learnt about the implementation of a complex exercise intervention for women with breast cancer, and the issues raised will inform the development of a future implementation strategy.

**Conclusions** A physiotherapist-delivered early supported exercise intervention with integrated behavioural strategies helped women at risk of shoulder problems following breast cancer treatment to feel more confident in their ability to mobilise their arm post-surgery. A physiotherapist-delivered early supported exercise intervention with integrated behavioural strategies may address the sense of powerlessness that many women experience during breast cancer treatment.

### Strengths and limitations of this study

► Interviewing multiple groups (intervention arm, control arm participants and physiotherapists) in this study enabled us to triangulate the data and explore experiences from multiple perspectives.

► We note that the participants we interviewed were a particularly motivated group, and it is possible we did not capture some of the challenges which other, less motivated, women may have experienced.

► We obtained consent to be approached for interview prior to randomisation, independent of treatment allocation, in an attempt to minimise bias. We tried to minimise the risk of social desirability bias by asking neutral questions and explaining there were no right or wrong answers.

► We used a convenience sampling approach, which is a potential weakness of this study as it may have resulted in a lack of diversity among participants.

► Our sample was overwhelmingly white, with only one of the participants identifying as another ethnic identity. Findings may not reflect the experiences of black, Asian and other minority ethnic groups.

**Trial registration number** ISRCTN35358984.

## INTRODUCTION

Treatment to the chest and axilla for breast cancer can result in upper body problems, such as reduced range of movement in the shoulder, muscle weakness, pain, lymphoedema and functional limitations.[1 2] These problems can impact on ability to carry out activities of daily living and may persist for many years after treatment.[1 2] Exercise in the acute phase following breast cancer surgery may improve shoulder function in women at high risk of shoulder problems.[1] Guidelines state that people diagnosed with breast cancer should be referred to physiotherapy when indicated,[3 4] however, in the UK National Health Service (NHS) this is not routine practice. There is a need for a proactive model of

care which encourages early exercise-based rehabilitation and provides physiotherapists with resources to inform their practice.[5]

Loss of a sense of control, loss of self-identity, and alienation from their bodies during and after treatment are often reported by individuals with cancer.[6–12] It has been proposed that improving women's self-efficacy through physical rehabilitation may improve their quality of life.[6] Lack of knowledge about exercise and the experience of cancer-related fatigue were identified by individuals with cancer as obstacles to exercise in a recent Korean study.[13] A recent systematic review of the qualitative literature identified six studies of mixed quality reporting the experiences of women living beyond breast cancer participating in a supervised exercise intervention.[14] These studies all reported on group interventions, and the findings suggest that the group element may be beneficial. Little is known about the experiences of this patient group participating in individual supported exercise intervention. There has also been little published regarding the experiences of professionals delivering exercise interventions to this patient group. One recent study reported a lack of confidence among physiotherapists in treating people with cancer, but respondents felt confidence grew with practice.[15] Little is known about how physiotherapists feel about the feasibility of implementing a service for people with breast cancer. This is important so that we can address challenges and issues when designing services.

The UK Prevention Of Shoulder Problems (PROSPER) trial evaluated the clinical and cost-effectiveness of an early supported home-based physiotherapist-led exercise intervention in women with newly diagnosed breast cancer at higher risk of developing shoulder problems after treatment.[16 17] We have published a description of the intervention and trial protocol elsewhere.[16 17] In this paper, we report the findings of the UK PROSPER trial embedded qualitative study.

The aims of the qualitative study were:

► To understand the acceptability of the exercise intervention to participants.
► To explore how the exercise intervention or control affected their experiences of recovery after cancer treatment.
► To investigate the experiences of physiotherapists delivering the exercise intervention.
► To explore participants' and physiotherapists' perspectives on issues related to the implementation of the PROSPER programme to inform future plans for implementation.

Figure 1 illustrates the pathway of participants through the trial and embedded qualitative study.

## METHODS
### Methodology
The study was underpinned by critical realism, assuming that an underlying reality is experienced and given meaning by individuals.[18 19] To meet the study aims, we conducted qualitative semistructured interviews with reflexive thematic analysis.[20] This allowed for exploration, depth and understanding of the experiences of trial participants, thus taking an interpretive 'sense-making' approach rather than hypothesis-testing or confirmatory approach. We used the Standards for Reporting Qualitative Research reporting guidelines checklist.[21]

### Sampling and recruitment
#### Trial participant interviews
On recruitment to the trial, we offered all trial participants the option to take part in an interview at a later date (see figure 1). We recorded signed consent to be approached for interview and this formed our sampling frame for the qualitative study. We approached women in the intervention arm after they were discharged from physiotherapy to avoid contamination bias. The researcher (SR) telephoned participants to invite them to interview; and if they expressed an interest, participants were sent an information sheet and interview consent form.

After conducting and analysing seven interviews with intervention participants, we decided to interview control arm participants, to compare their experiences. We used our database of those who had consented prerandomisation to select a sample comparable with the intervention sample in terms of time since randomisation, so that women were at similar stages of postoperative treatment and could reflect back over their experiences of recovery.

#### Physiotherapist interviews
We informed all physiotherapists delivering the intervention about the interview study. We then sampled physiotherapists from low and high recruiting trial sites to allow exploration of different perspectives on intervention delivery. SR approached therapists via email or telephone.

### Data collection
We developed flexible topic guides with women with breast cancer, based on the aims of the study and relevant literature. One-off interviews were conducted by SR either at the participant's home, by telephone (trial participants) or in a private room at their place of work (physiotherapists). Physiotherapists who worked together were interviewed in pairs. The physiotherapists who volunteered for the interviews and were interviewed in pairs worked closely together. Interviewing them in pairs allowed physiotherapists to share and reflect on their experiences and aided recall where they had only treated a small number of participants, for example. It is possible that interviewing them in pairs could have affected their responses, but participants were remarkably candid about the challenges they experienced, thus we were not concerned that this was happening. Only the researcher and interviewees were present. All interviews were audio-recorded. We took study materials (physiotherapy manual, participant materials) into the interview

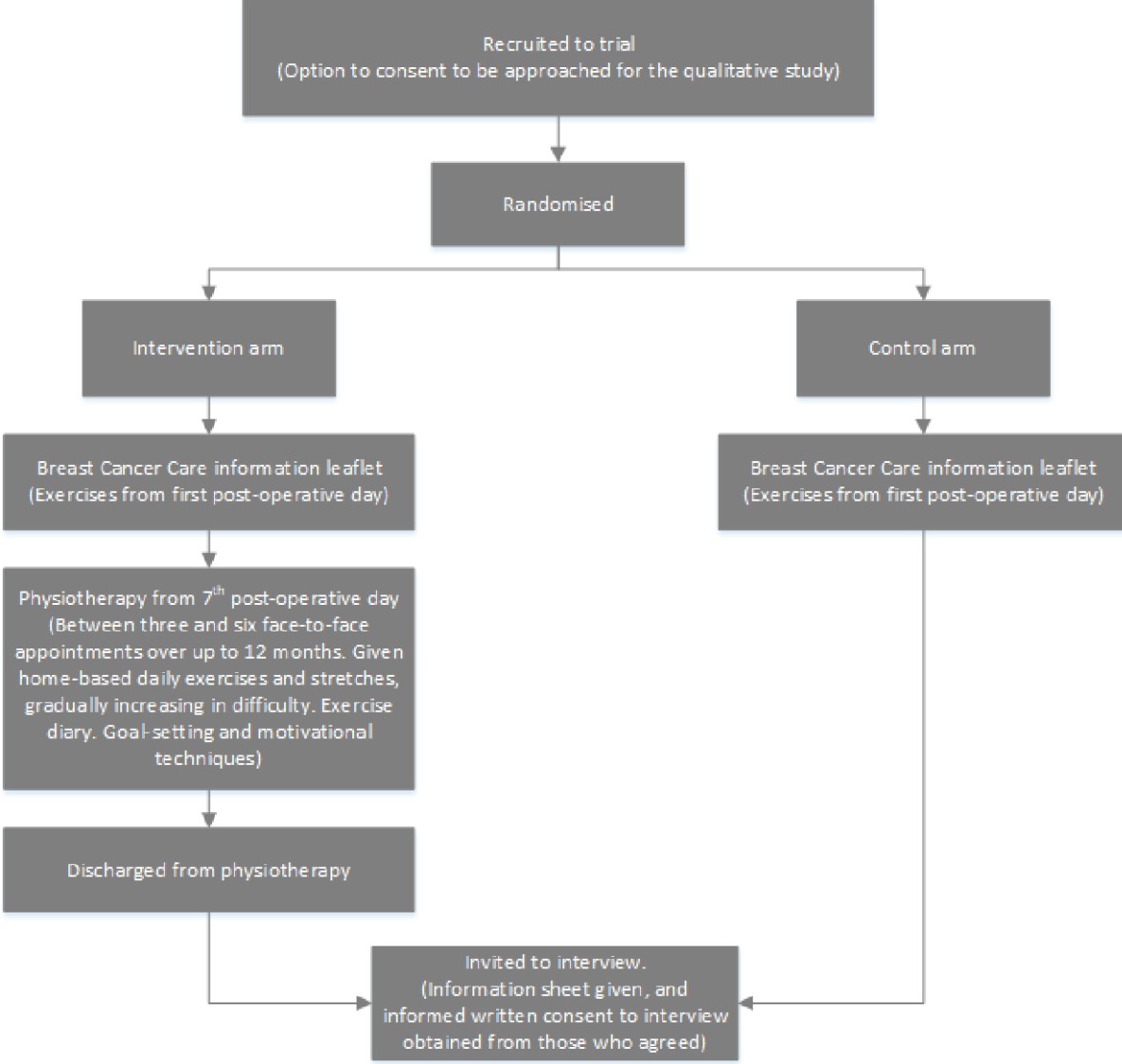

**Figure 1** Participant pathway through trial and qualitative study.

to aid recall and discussion. Informed consent was gained before the interview began.

**Data analysis**

Interviews were transcribed verbatim, checked for accuracy and anonymity by SR, and then uploaded to QSR NVivo Pro V.11.[22] Thematic analysis[23 24] was conducted by SR and managed in NVivo. Interview transcripts were 'coded', where sections of text are assigned a descriptive label, producing dozens of codes per interview. These codes were then grouped into categories, and these were then grouped further into themes. Analysis began alongside data collection. As a research team, we met regularly to discuss emerging findings and the evolving analysis.[25] Saturation in this study meant that we had enough data to understand each of the identified categories and themes, rather than that there was 'nothing new' to be found.[26 27] We reached saturation after 15 trial participant interviews and 5 physiotherapist interviews.

**Reflexivity and rigour**

Interviews were conducted sensitively by a female researcher experienced in interviewing people with cancer (SR).[12 28–31] The evolving analysis was discussed with the research team (SR/JB/HR/BM). SR is a social scientist with expertise in qualitative research with people with health conditions, including breast cancer, and healthcare professionals. HR and BM are researchers and physiotherapists. JB is a trialist and PROSPER chief investigator, she did not influence the qualitative study findings, but provided important contextual details regarding the trial and intervention. We were careful to conduct balanced interviews, without assuming that the trial participants and physiotherapists would have positive views of the intervention. SR reminded interviewees throughout that she was not involved in the development of the intervention and welcomed their honest views. Rigour was assessed using Lincoln and Guba's conceptualisation of trustworthiness.[32] SR collected the data and

**Table 1** Study sample (trial participants)

| Characteristic | Intervention arm N=10 | Control arm N=10 |
|---|---|---|
| Months since randomisation, mean (range) | 7 (3–11) | 7 (3–12) |
| Age at randomisation, mean (range) | 51 (28–69) | 60 (44–79) |
| Age at randomisation | | |
| 18–29 | 1 | 0 |
| 30–39 | 0 | 0 |
| 40–49 | 4 | 2 |
| 50–59 | 3 | 3 |
| 60–69 | 2 | 3 |
| 70–79 | 0 | 2 |
| Ethnicity | | |
| White | 9 | 10 |
| Mixed | 1 | 0 |
| Surgical treatment* | | |
| Mastectomy | 4 | 3 |
| Breast-conserving surgery | 6 | 7 |
| Axillary node clearance | 10 | 8 |
| Sentinel lymph node biopsy | 3 | 4 |
| Adjuvant therapy* | | |
| Chemotherapy | 9 | 7 |
| Radiotherapy | 9 | 10 |

*Participants had multiple treatments.

was immersed in the data during analysis. Quotes have been provided to illustrate themes.

## RESULTS
### Sample
We recruited 392 women (196 per arm) to the PROSPER clinical trial from 17 breast cancer centres in England. Overall, 67% (n=264/392) of trial participants provided signed consent to be contacted for an interview. In total, we attempted to contact 53 women regarding an interview. Of these, 11 were not contactable, 17 agreed initially for an interview but could not be reached again and 5 declined. Ten participants from the intervention arm and 10 from the control arm were interviewed from 11 of the 17 study sites (see table 1). There were no apparent differences between the sites regarding the issues raised by trial participants and therapists. We had a good range in terms of size and rural/urban sites across the 11 sites represented in the interview study.

Interviews were carried out with 11/44 (25%) physiotherapists (all women) from six study sites. Ten were interviewed in pairs and one individually. The physiotherapists had treated between 1 and 16 trial participants (median 5) and were based at hospitals that did

not routinely provide postoperative physiotherapy after breast cancer surgery. They were experienced in the management of musculoskeletal conditions but did not currently work in breast cancer or oncology units. Some physiotherapists had experience of treating people with breast cancer presenting with problems such as restricted shoulder movement preventing the start of radiotherapy. One physiotherapist had past experience working on a cancer inpatient ward.

We identified three themes from the data: 'healing'; 'being the perfect therapist' and 'delivering physiotherapy to women with breast cancer'. The themes and subthemes are illustrated by participant group in table 2, and each theme is described below with subthemes.

### Healing
This theme refers to trial participants' and physiotherapists' comments about how the exercise intervention shaped the experience of healing for the women with breast cancer.

#### Reassurance
In the acute period after surgery, participants from both the intervention and control groups reported feeling afraid to move their upper body. This is known as kinesiophobia.[33 34] They felt unable to do the exercises prescribed in the Breast Cancer Care information leaflet.

> It's quite tender…you don't feel like you ought to be doing it…you feel like it's too soon…I was aching so much that I just thought 'I just can't do this'. —Qualitative Respondent (QR24) (age 51, control arm participant)

> Because you don't know whether it's good or not, do you know what I mean, you don't know if you're doing well or not or if this is where you would be, you know, or you should be and it was quite nice because you got 'oh no you're doing really well' or 'oh yeah that will be tight' and it was that… it was quite nice to have the feedback. —QR12 (age 55, intervention arm participant)

Participants allocated to the intervention arm subsequently felt reassured by physiotherapists that they were capable and able to move. They felt reassured that bodily sensations, such as stiffness, were normal and not something to worry about. Physiotherapists felt that they were able to increase participants' confidence in moving their bodies, and that this lifted participants' confidence more broadly.

> The, er, physiotherapist…was able to tell you whether you were doing things right or wrong or how things were going within your body. —QR09 (age 69, intervention arm participant)

> Interviewer: What do you think they get out of coming to see you?

> PT02 (Physiotherapist): I think confidence. Confidence to actually move…confidence to look after themselves, that they can do things.

**Table 2** Illustration of themes and subthemes by participant group

| Theme | Subtheme | Intervention group | Control group | Physiotherapists |
|---|---|---|---|---|
| Healing | Reassurance | The physiotherapist was able to tell you whether you were doing things right or wrong | It's quite tender…you don't feel like you ought to be doing it…you feel like it's too soon | They think they're going to split their stitches…it's just the reassurance we give |
| | Making progress | You saw results and sometimes with your cancer…you don't see results until the end | I'm tender but I suppose that will go | |
| | Helping myself | I think it was because you were doing something, because so much of cancer care is being done to you | It still gets stiff now but you just have to deal with it. | They had the confidence to do it for themselves |
| | Looking ahead | Now I'm just doing the massage for lymphoedema and exercises only if I feel the problem | | |
| Being the 'perfect' therapist | | She asked me how I felt and it was very much about me and my progress | | It almost like it made you be the perfect physio and the perfect way you should treat patients but you don't always have time to do that |
| Delivering physiotherapy to women with breast cancer | Meeting the needs of women with breast cancer | I stopped doing the exercises for three or four days if I was ill and after that it was more difficult to do the exercises after | | They'd start their chemotherapy and then it was a whole different ball game because it was just kind of managing their fatigue and we struggled to get people back in for appointments |
| | Emotional support for physiotherapists | | | There were times where it was upsetting to hear |
| | Physiotherapists' time, skills and organisational integration | | | I would say giving them the choice of exercise is time consuming which you wouldn't have in real life, you wouldn't have the time<br>There's not necessarily the integration with like the nurses or the lymphoedema team, we are quite a separate team<br>Being able to advise people a bit more around like scar massage or kind of…any of the manual treatments that we could've done and when is right and wrong to use them [was difficult] |

Some described this as giving participants 'permission' to move, which was necessary to prevent movement restrictions in the upper body.

### Making progress

This theme refers to physical improvements felt by participants in the intervention arm. This included how far they could stretch and how strong they were. Improvements were measurable and tangible, and participants highlighted the central role of the physiotherapist in creating this sense of progress.

You saw results and sometimes with your cancer…you don't see results until the end 'til they say 'You're all clear' you are just going through awful, awful, awful praying and hoping…But it is a really positive thing to think 'Oh something is getting better'. —QR12 (age 55, intervention arm participant, participant's emphasis)

You could kind of measure it yourself and assess it yourself because you knew how far you could get your arm up…You could feel when, when things started

to get a bit better. —QR08 (age 43, intervention arm participant)

Where intervention participants spoke about the improvement they felt in the months following their surgery, control arm participants also spoke about improvement, but for them this remained an ongoing process even more than 12 months on.

When I'm washing myself and, and if I touch myself I'm tender but I suppose that will go. —QR17 (age 67, control arm participant)

I have still got a seroma on my chest which is a bit of a nuisance which is, um, sort of swelling of fluid isn't it. Um, it's less than it was and I think it's gradually going 'cause it was enormous at the very beginning but it's getting less. —QR16 (age 79, control arm participant)

Over time, intervention participants progressed from gentle stretching to more advanced stretching and strengthening exercises as they improved. Graduating to harder exercises gave them a sense of achievement.

When we would do the exercises and when we would move the kind of categories in the folder that was given, that made me feel good and made me want to kind of continue. —QR13 (age 28, intervention arm participant)

Progression was fulfilling and rewarding, particularly in the context of cancer treatment where a sense that they were improving or getting better was lacking. To be able to measurably perceive progress in strength and movement helped to restore a sense of bodily autonomy for the women who felt disempowered by cancer treatment. During this profoundly difficult time of undergoing cancer treatment, feeling improvement and graduating to harder exercises helped them to feel that they were getting better, at least in some way.

### Helping myself
During breast cancer treatment, women passively receive treatment.[7–9 35] One participant described it as being 'a professional waiter, you just sit and wait, and you just let everyone do what they're doing' —QR23 (age 62, control arm participant).

In collaboration with their physiotherapist, participants receiving the exercise intervention could choose which exercises they performed from a menu, selecting exercises they felt most confident and comfortable doing. The physiotherapists felt that joint decision-making was a patient-centred approach which added to trial participants' sense of ownership and control of their exercises. Physiotherapists and participants both noted that this gave participants an opportunity to be proactive, taking control of one aspect of their recovery. This sense of responsibility and ownership motivated them to exercise.

I think it was more than the exercise. I think it was because you were doing something, because so much

of um cancer care is being done to you…It was just quite nice to have something proactive for you to do rather than just turn up and have the drugs. —QR12 (age 55, intervention arm participant)

I think when we were sort of promoting why we think the exercises were useful I talked about self-determination this is something that you can do for yourself and your care…particularly the way it was designed that enabled the patients to say well we could do this exercise or we could do that one. —PT10

This was a key difference to the control arm interviewees. Most control arm participants spoke about accepting postoperative problems or just waiting for them to improve over time, apart from a few highly motivated individuals who described inventing their own exercises.

Lifting up now and I can feel the stretching down that left-hand side, but, um, you know I don't know, I suppose it's had trauma. —QR15 (age 44, control arm participant)

The tightness on my chest does limit movement sometimes and it's sort of more of a discomfort than a pain and just a blessed nuisance really but everyone I've seen says it's normal that they take a while and it's nothing they can do so it will just go when it's ready I suppose and I kind of live with it. —QR17 (age 67, control arm participant)

Wanting to be a 'good patient' and doing as one was told motivated control group participants to follow the exercises on the Breast Cancer Care leaflet.

Well I'd like to think I was a good patient, I started my exercises the day after I came out of hospital. —QR26 (age 56, control arm participant)

Just the fact that the hospital gave them you and, you know, they know what they're talking about. You do it because you've been told to. —QR23 (age 62, control arm participant)

This is in contrast to the sense of self-determination, control and progress described by participants receiving the intervention.

### Looking ahead
Some participants continued to draw on knowledge gained from the intervention to alleviate ongoing problems with tightness and stiffness, and appeared to feel quite confident in managing this in the future.

Now I'm just doing the massage for lymphoedema and exercises only if I feel the problem…For example if, if I feel the problem to reach the shelf I'm taking [the] band and I might warm it up just do the exercises with the elastic band exactly for this movement. —QR10 (age 50, intervention arm participant)

In a couple of months or so, I would like to kind of start using weights so that I can strengthen my arms… It's kind of like building up the strength that I was

building towards whilst I was doing the [PROSPER] exercises before. —QR13 (age 28, intervention arm participant)

They felt assured that continuing with such activities would help them, and that they would know what to do or where to seek help if required.

### Being the 'perfect' therapist

This theme describes the physiotherapists' perspectives on the trial intervention compared with their usual practice and how it enabled them to deliver an optimal service.

It almost like it made you be the perfect physio and the perfect way you should treat patients but you don't always have time to do that. —PT03

Agreed goals [and] agreed exercises actually that should be what we're doing anyway that shouldn't be anything radically different but sometimes because of time pressures you don't…If you work more collaboratively with patients there are massive benefits to it and I think it just reinforced that for me. —PT10

This was supported by participant responses, where they described the relationship they built with their physiotherapist.

For me, you know, having the same… the same desired outcome as the physiotherapist and [wife] you know, kind of, being all, all, all wanting the same thing. And it kind of felt if I did those things then I would eventually achieve it. —QR08 (age 43, intervention arm participant)

She's brilliant, she's so lovely and fantastic hugger that's what I found if somebody you meet is happy to give you a hug when you are in this kind of situation it… it just makes everything so much easier… you [physiotherapists] not only do you do your jobs but when you're dealing with people like me you are counsellors as well. —QR12 (age 55, intervention arm participant)

Physiotherapists felt that having longer appointment times and an emphasis on shared goals and shared decisions, both of which encouraged exercise adherence, represented an ideal way of working. Many of the therapists remarked that they were pleased to be able to offer this service to people with breast cancer because of their previous experience of treating women struggling with chronic immobility, pain and psychological issues in the longer term as a result of shoulder problems following breast cancer surgery.

When you pick up those patients [later] they come with a lot of emotional baggage and sort of their belief systems and it may have been years since they used their shoulder normally and then you know again if you've got body dysmorphic issues and they've been carrying that around for two years that's a lot more challenging to support. —PT06

We get people coming in about two years later and they've never touched their scar, they never saw a physio, they're stiff, their scar's horrible, they've got awful myofascial trigger points and tightness… They still think two years down the line they're going to hurt themselves if they over stretch so if you get them in at the early stage then it's just better…I had a lady who had a mastectomy it was three years later she never went back to work, she never went back to any exercise, she never touched her scar, her mental wellbeing was like absolutely awful when I first started seeing her because she just didn't even know that she could have her life back. —PT01

Physiotherapists connected this to the broader organisation of the NHS, and the need allocate resources to preventive care.

I think we work too much reactive in the NHS don't we and I think a direction to move in is work in prevention rather than cure. —PT02

### Delivering physiotherapy to women with breast cancer

This theme reports views on delivering a new physiotherapist-led intervention for individuals with breast cancer.

#### Meeting the needs of women with breast cancer

Participants and physiotherapists suggested that adjuvant treatment, such as chemotherapy, interfered with the participants' ability to maintain the exercise programme. After stopping the exercises when they felt unwell, it was physically more difficult to start doing the exercises again. Physiotherapists reflected that intervening at this point may have helped encourage and motivate participants to continue.

On day 17 after chemotherapy it has been a struggle…the last three weeks with the first lot of chemo this…[doing the exercises has] been a lot harder than I ever anticipated. —QR11 (age 49, intervention arm participant)

A patient would come in for their first appointment and probably just post-surgery and most of them were quite positive had quite a lot of goals… they'd start their chemotherapy and then it was a whole different ball game because it was just kind of managing their fatigue and we struggled to get people back in for appointments. —PT02

The physiotherapists noted that participants needed emotional support, and that it was difficult to provide this in a curtained cubicle in an open-plan space, where they potentially felt more vulnerable.

We're working in curtained cubicles a lot of the time and I felt that didn't set the tone, I think if you're asking someone to take their bra off. —PT10

Two therapists felt that physiotherapists should be women as they would better understand the meaning of

losing a breast and more able to engage in the emotional and physical work of treating women with breast cancer.

> They would probably connect better with a female and I was surprised how much women wanted to talk to me about their connection with their breasts so for a lot of them they felt like that was their femininity or that was um a connection to their womanhood and so I think most guys couldn't relate to how that feels so I could get where they were coming from. —PT08

This issue was not mentioned by the trial participants we interviewed.

### Emotional support for physiotherapists

Physiotherapists typically provide emotional support to their patients, however, some therapists highlighted particular challenges in relation to this group due to the context of cancer treatment, for example, people with cancer were fearful of dying from breast cancer. This was in contrast to their usual caseload which often involved caring for people with chronic musculoskeletal conditions.

> I am a person who cries quite easily so I was like 'Ok I need to keep things under control myself because I am the professional'… If I was to do it longer term I would need some better kind of guidance and help to deal with that…sometimes I felt a little bit lost. —PT08

> We were lucky because we had each other but there were times where it was upsetting to hear…If we were permanent members of staff in oncology you would be given some…de-briefing or kind of decompression but we were never offered that… both of us have had very close relatives die because of cancer…nobody considered that at all. —PT06

The physiotherapists felt they would need emotional support if they worked routinely with people with breast cancer.

### Physiotherapists' time, skills and organisational integration

Delivering the intervention was time-consuming for physiotherapists.

> I would say giving them the choice of exercise is time consuming which you wouldn't have in real life, you wouldn't have the time. —PT09

Trial appointments were longer than usual and there were doubts about how this could be practically implemented as part of routine NHS clinical care given current time restrictions on appointments. The physiotherapists felt confident in identifying and treating physical shoulder problems, but often expressed a need for training about breast cancer, its treatments and cancer-specific complications. Cording, lymphoedema and seroma were unfamiliar postoperative complications to some physiotherapists until they took part in the trial.

> We are MSK [musculoskeletal] physios and we know what a tight shoulder is and we know how to get it moving, so actually the assessment and the exercises wasn't so much of a worry, but patients occasionally asked me a question that maybe I couldn't answer… the background behind the cancer, a bit more about the actual surgical techniques they did and why and a little bit more about the reasoning of why lymphedema and cording does actually develop and what it means, I might have benefitted from more training from that aspect. —PT03

> Being able to advise people a bit more around like scar massage or kind of…any of the manual treatments that we could've done and when is right and wrong to use them [was difficult]. —PT02

Physiotherapists felt disconnected from the surgical or oncology team treating the person with breast cancer which was challenging.

> There's not necessarily the integration with like the nurses or the lymphedema team, we are quite a separate team from them so I think it does need to be a multi-disciplinary approach and because we're not involved with them it makes it a little bit difficult [to know] whether we should or shouldn't be doing those interventions. —PT03

> I sometimes found it difficult to ask about things like chemo, radiotherapy and repeat surgeries because I almost felt like it was something that I should know… that's what I'd want as a healthcare professional I want them to know what's going on I shouldn't have to tell you when I am having my chemo or this is happening. —PT05

Better integration with the oncology team would have given them greater understanding of the specific individual's treatment schedule as they sometimes felt uncertain about whether the interventions were appropriate at a particular stage of cancer treatment.

## DISCUSSION

This qualitative study embedded within a large multicentre clinical trial makes a unique contribution to the literature. Our study illustrates that an individual supported exercise intervention is perceived as acceptable and beneficial by both women with breast cancer and physiotherapists. Comparing the intervention and control arm enabled us to demonstrate that the intervention helped participants feel empowered and regain a sense of control, whereas participants in the control arm spoke of passively accepting the upper limb limitations they experienced. Previous studies have explored perceptions of exercise in the context of an exercise intervention,[36–38] but this is the first to include the perspectives of both intervention and control group participants, as well as physiotherapists delivering the intervention. We gained multiple perspectives on the same issue and included all

stakeholders in the study. This allowed us to triangulate and identify themes which were present across all groups. By using qualitative methods, we elicited the particular elements of the intervention which helped motivate participants, and those which were easier or more difficult to deliver in the clinical setting. This intervention is the first early structured physiotherapy-led home-based exercise intervention to be tested in women with breast cancer in the UK. An understanding of the acceptability of the intervention to women with breast cancer and physiotherapists will inform future implementation strategies if the intervention is clinical and cost-effective.

Uncertainty has been identified as a feature of the experience of cancer.[29 39] The subtheme of 'making progress' showed how witnessing improvement for themselves in terms of strength and stretching stood out in sharp contrast with the uncertainty surrounding cancer and its treatment. Participants also gained a sense of control over their progress, through being involved in choosing exercises and through taking responsibility for completing their exercises each day (subthemes of 'helping myself' and looking ahead'). This combination appeared to restore participants' sense of autonomy over their bodies, and improved their well-being as they felt less disempowered and hopeless. This echoes previous research which found feelings of increased empowerment when people with breast cancer participated in physical activity during active treatment.[40–42] These experiences contrasted to those in the control group, who did not experience the same sense of empowerment and progress. Specific aspects of the intervention which contributed to this sense of control over and above usual care were the contact with physiotherapists and the reassurance this provided, the sense of progress working through the prescribed programme as exercises increased in difficulty and the shared decision-making used to select the home exercises. Previous research has found that participating in a group activity can be a way of forgetting about the illness.[36] Our study illustrates this can also be true for home-based or individually supported exercise programmes.

Being diagnosed with a serious illness such as cancer can cause an individual to lose trust and confidence in their bodily knowledge and of what their bodies are capable of doing.[12 43–45] The women in this study reported kinesiophobia (fear of movement) in the acute period following surgery, but those in the intervention arm felt the intervention helped them overcome this (subtheme of 'reassurance'). Kinesiophobia has been shown to be associated with lymphoedema and greater pain intensity.[33 34] Physiotherapists were able to reassure women that their bodily sensations were normal and gave them confidence to push themselves physically, which motivated them to adhere to the programme. The interview data suggested that the role of the physiotherapist in affirming this progress and confidence was crucial. Physiotherapists provided invaluable emotional support, as participants unburdened onto them and shared their fears about the future and their bodies.

The physiotherapists enjoyed seeing positive improvements in the participants and felt passionate about delivering what they viewed as high-quality care to individuals with breast cancer (subtheme of 'being the perfect therapist'). Physiotherapists felt satisfaction in being able to take preventive action against problems arising in the future for these women. Intervention arm participants also appreciated the supportive nature of the intervention, sharing decisions and working together towards the same goal. Other authors have called for a more proactive model of healthcare provision for this patient group and identified the need to improve physiotherapists' confidence in supporting those with breast cancer.[46] Our physiotherapists felt that they needed better integration with the rest of the individual's healthcare providers (final subtheme of 'physiotherapists' time, skills and organisational integration'). Other studies have also emphasised the importance of aligning expectations and knowledge about exercise-based rehabilitation across the whole cancer care team.[5] Challenges to the exercise programme were the side effects of treatment, in particular fatigue, which has been highlighted in other research as a barrier to exercise for people with breast cancer.[36] If a physiotherapist can provide motivation and encouragement during chemotherapy, this may improve adherence to exercise. However, it is also important to ensure that physiotherapists are sympathetic to treatment-related issues and can tailor programmes during these periods of fatigue.[36]

The theme 'Delivering physiotherapy to women with breast cancer' highlighted considerations for the implementation of an exercise intervention for patients with breast cancer. The intervention should be delivered in a private walled room, ideally with a specifically trained (female) physiotherapist who is part of the multidisciplinary oncology team caring for the person. The most important ingredient of the intervention was contact with the physiotherapists, suggesting resources should be focused on training and supporting physiotherapists to provide this care. Some physiotherapists reported feeling upset when treating patients because of the woman's distress or their own experiences of cancer. This suggests that healthcare professionals caring for oncology patients should be given the opportunity of debriefing and emotional support. This is an important consideration when designing future interventions for this group.

In the PROSPER trial, participants underwent a 1-hour assessment and then subsequent 30-minute follow-up appointments. Routine UK physiotherapy outpatient appointments are often 40 min (assessment) and 20 min (follow-up). Physiotherapists worked with participants to select the exercises. This may be challenging to deliver in a resource-stretched NHS context. However, longer appointments with physiotherapists, creating shared goals and making shared decisions about exercises were viewed as the most important ingredients in the successful delivery of the intervention. This was brought out in all our themes. Other studies have highlighted how autonomy of choice over exercises may increase

motivation and adherence.[37 47] Additionally, we provided the PROSPER materials in an attractive ring binder with colour photographs and laminated sheets, and provided exercise diary handouts. Trial participants said the diary was useful as a prompt to remember to do their exercises, and it was helpful to see photographs of the exercises.

Acting more proactively by providing good access to physiotherapy treatment early after, or alongside, breast cancer treatment could help to reduce the number of people with cancer (or a history of cancer) presenting with musculoskeletal complications.[48] Although our physiotherapist-participants felt very comfortable with aspects of the intervention such as improving shoulder mobility, they expressed a need for greater training, support and guidance in relation to specific issues such as cording and lymphoedema. The physiotherapists delivering the PROSPER intervention were musculo-skeletal specialists with limited experience in treating individuals with breast cancer in the acute postoperative period. Physiotherapists in the UK receive little training in rehabilitation following cancer treatment, reflected by the limited centres across the UK with physiotherapists specialised in oncology.[48] Given the increasing number of people surviving cancer and living with the consequences of cancer treatment, there is an urgent need in the UK to upskill physiotherapists in cancer-related rehabilitation to allow people with breast cancer better access to this type of rehabilitation.

## CONCLUSION

This study has highlighted how a physiotherapist-led home exercise programme, with built-in progression and shared decision-making, helped women undergoing breast cancer treatment gain a restored sense of control over their well-being and empowered them during a highly disempowering experience.

**Acknowledgements** We would like to thank all the physiotherapists and trial participants who took part in this interview study. We dedicate this paper to Elizabeth (Lizzie) Abbey (nee Fort) who passed away on 3rd April 2018. Lizzie helped set up the trial and was a highly valued member of the PROSPER team.

**Contributors** JB is chief investigator of the PROSPER trial. JB/SR/EW/HR/BM contributed to the study protocol. SR collected and analysed the data. JB/BM/HR/EW assisted with analysis and interpretation. SR led the writing of the paper, and all other authors contributed to writing and editing.

**Funding** Funding for the PROSPER trial was provided by the Health Technology Assessment programme of the National Institute for Health Research (NIHR) (project number 13/84/10). JB is supported by NIHR Research Capability Funding via University Hospitals Coventry and Warwickshire. EW is supported by the NIHR Applied Research Collaboration Oxford and Thames Valley.

**Competing interests** None declared.

**Patient and public involvement** Patients partnered with us for the design of the study, the informational material to support the qualitative research and the burden of the interview from the perspective of women with breast cancer.

**Patient consent for publication** Not required.

**Ethics approval** The study was approved by the National Research Ethics Service Committee West Midlands Solihull on 20th July 2015 (ref no: 15/WM/0224).

**Provenance and peer review** Not commissioned; externally peer reviewed.

**Data availability statement** No data are available. The datasets generated during and/or analysed during the current study are not publicly available due to the need to protect the identity of participants.

**ORCID iDs**
Sophie Rees http://orcid.org/0000-0003-4399-2049
Bruno Mazuquin http://orcid.org/0000-0003-1566-9551
Esther Williamson http://orcid.org/0000-0003-0638-0406
Julie Bruce http://orcid.org/0000-0002-8462-7999

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
