## [Reviewer comments · BMJ Open]

ARTICLE DETAILS

TITLE (PROVISIONAL)	The role of physiotherapy in supporting recovery from breast cancer treatment: A qualitative study embedded within the UK PROSPER trial
AUTHORS	Rees, Sophie; Mazuquin, Bruno; Richmond, Helen; Williamson, Esther; Bruce, Julie; -, UK PROSPER Study Group

VERSION 1 – REVIEW

REVIEWER	Amy Dennett Eastern Health Australia
REVIEW RETURNED	16-Jun-2020

GENERAL COMMENTS	Abstract: include how participants were sampled Introduction: Limited background information provided. Need to provide further information about where the current gap in literature is. What other qualitative studies have been done if any? If an aim is to explore future implementation, include in the introduction what and why this is important. Results: Further clarity and information about any similarities or difference between patients and therapists would add more depth to the results. In addition, the theme relating to being the 'perfect therapist' appears to relate to just the therapist perceptions and is much less than the other themes. Therefore, rather than being a main theme may be more a subtheme. Or, was there any discussion about patients and how they valued staff as it is known in other areas of cancer rehabilitation staff patient relationships are important. Overall, the results could be presented more clearly. A figure depicting the relationship of themes would be helpful. Differences between the control and intervention groups could also be more clearly highlighted You have also mentioned triangulation in the methods but this is not discussed in the results. Please include. Discussion Summarise the main findings in the first paragraph of the discussion. Really highlight what this study adds Page 16 line 37 – use person first language ie people with breast cancer and use throughout manuscript Consider discussing findings and implementation in relation to an implementation framework Further discussion of limitations required Conclusion could be stronger. Add how the PT led program helped wellbeing
---

REVIEWER	Jenna Smith-Turchyn McMaster University, Canada
-----------------	--

GENERAL COMMENTS

Overall, this is a very well written paper and provides interesting and useful information on an important topic. Thank you for considering the qualitative components alongside your quantitative trail. Updates to a few sections would further strengthen this paper and the usefulness of the information for readers. Overall, I think this is an important topic for the PT profession and helps to demonstrate the need for PTs as part of the cancer care team.

Methods: Data collection

- Can more information be provided on the interview questions asked to participants and physiotherapists? If not specific questions, What did they relate to? How were they framed? Could this be provided in text or in a table or as an appendix for interested readers?

Methods: Thematic analysis

- Describe in more detail for readers (summarize in a few lines only).
- Overall methodology sound and good detail provided within manuscript.

Results:

- Did anything differ between sites of those that were interviewed compared to those that were not interviewed? Could their experience be different? Perhaps relate to size, location, etc. to contextualize for reader
- Why did you decide to interview PT's in pairs? This point should be in the methods section and described in more detail. Could this have affected the responses of PT's?
- Can you give more information on the PT's interviewed in chart as you did with the patient participants (age, years treating, number with previous oncology experience, etc.). This will help link to your discussion of PTs previous experience with cancer patients and how this may have affected delivery of intervention.
- **Can you embed quotations within all parts the results section?** I always find this brings more meaning and understanding to the results. Also, link to participant (age, stage, etc.) and can analyze results based on some of these factors. Doing this once or twice within each section and then referring to table would improve this paper (like did in "Helping Myself" section)
- Try and summarize results more clearly based on intervention group; this was done well in some sections but was not clear in others.

	 • Can the results be summarized in a figure / chart comparing group perhaps or integrating some of the subthemes? Or comparing multiple perspectives. As you have highlighted, I think this is a strength of this paper, but it could be presented more effectively in the results. • Link results back to objective of acceptability; other aims clearly linked, but this aim should be more clearly addressed for readers. Also, how is acceptability defined, how do results link, etc. Discussion:  • Can touch on / explore more training of these physiotherapists. Described in good detail how they felt if working with this population was new. Now, what additional training is needed? Did they have any MI training as well? If yes / no, why might this be important based on your results / discussion points? • At times I feel like new ideas around results are presented in the discussion. Link points to themes more clearly.
--	---

VERSION 1 – AUTHOR RESPONSE

Review	Reviewer comments	Revisions made
1	Abstract: include how participants were sampled	We have added 'Trial participants were sampled using convenience sampling. Physiotherapists were purposively sampled from high and low recruiting sites.'
1	Introduction: Need to provide further information about where the current gap in literature is. What other qualitative studies have been done if any?	At the time of funding in 2014, there were few studies in the UK setting. We have added more information about existing research to the introduction.
1	If an aim is to explore future implementation, include in the introduction what and why this is important.	We have added: 'Little is known about how physiotherapists feel about the feasibility of implementing a service for people with breast cancer. This is important so that we can address challenges and issues when designing services.'
1	The theme relating to being the 'perfect therapist' appears to relate to just the therapist perceptions and is much less	We appreciate this comment and have reflected. In response we have expanded this theme to make it stronger and to bring in the patient perceptions. Although we think these were included in earlier

	than the other themes. Therefore, rather than being a main theme may be more a subtheme. Or, was there any discussion about patients and how they valued staff as it is known in other areas of cancer rehabilitation staff patient relationships are important.	themes (for example, the importance of goal-setting to measure progress), we have included quotes where patients explicitly spoke about their relationship with their physiotherapist to strengthen this theme.
1 and 2	Further clarity and information about any similarities or difference between patients and therapists would add more depth to the results. Overall, the results could be presented more clearly. A figure depicting the relationship of themes would be helpful. Can the results be summarized in a figure / chart comparing group perhaps or integrating some of the subthemes? Or comparing multiple perspectives. As you have highlighted, I think this is a strength of this paper, but it could be presented more effectively in the results.	We have created a table depicting the themes and subthemes to illustrate and summarise from each group
1	Differences between the control and intervention groups could also be more clearly highlighted.	We have edited the results section now and hope it more clearly identifies the similarities/differences between the groups
1	You have also mentioned triangulation in the methods but this is not discussed in the results. Please include.	We have added the following sentence: 'This allowed us to triangulate and identify themes which were present across all groups.'
1	Summarise the main findings in the first paragraph of the discussion. Really highlight what this study adds	We have added the following sentence to the first paragraph of the discussion: 'Our study illustrates that an individual supported exercise intervention is perceived as beneficial by both patients and physiotherapists. Comparing the intervention and control arm enabled us to demonstrate that the intervention helped participants feel empowered and regaining a sense of control, whereas control participants spoke of passively accepting the upper limb limitations they experienced.'
1	Page 16 line 37 – use person first language ie people with breast cancer and use throughout manuscript	We appreciate this comment. We have chosen not to use the term 'survivor' as a result of our own and others' previous research demonstrating that survivor can be a damaging term to some people with a history of cancer. The use of 'patients' here aids the paper's readability and is also accurate as we are speaking of people undergoing physiotherapy treatment.

1	Consider discussing findings and implementation in relation to an implementation framework.	Thank you for this suggestion. We have clarified the objectives of this study related to future implementation: exploring issues that would need to be considered in a future implementation strategy rather than identifying a strategy at this stage. This is ongoing work which will be informed by the issues identified in this study and an implementation framework has not yet been developed.
1	Further discussion of limitations required	As per the journal guidance, the strengths and limitations were included in the article summary. We have however added the following sentences:  - 'Findings may not reflect the experiences of black, Asian and other minority ethnic groups. - We tried to minimise the risk of social desirability bias by asking neutral questions and explaining there were no right or wrong answers'
1	Conclusion could be stronger.	Thank you, we have amended to read: 'This study has highlighted how a physiotherapist-led home exercise programme, with built-in progression and shared decision-making, helped women undergoing breast cancer treatment gain a restored sense of control over their wellbeing, and empowered them during a highly disempowering experience.'
1	Add how the PT led program helped wellbeing	This has been highlighted in the conclusion as described above.
2	Thank you for completing this important work. Overall, this is a very well written paper and provides interesting and useful information on an important topic. Thank you for considering the qualitative components alongside your quantitative trial. Updates to a few sections would further strengthen this paper and the usefulness of the information for readers. Overall, I think this is an important topic for the PT profession and helps to demonstrate the need for PTs as part of the cancer care team.	Thank you for your kind comments regarding our work.
2	Can more information be provided on the interview questions asked to participants and physiotherapists? If not specific questions, What did they relate to? How were they framed? Could this be provided in text or in a table or as an appendix for interested readers?	We would be happy to provide an upload of the topic guide used in the semi-structured interviews as a supplementary file. The topic guide was flexible and was used as a prompt rather than as a strict interview schedule. We opened interviews with a broad question to ask physiotherapists to talk about what it was like for them being part of the Prosper study. Follow-up questions

		were used to probe their experience of treating cancer patients (e.g. how it differs/is similar to their usual caseload, any gaps in their knowledge/training). We asked them about specific components of the intervention (we took the physiotherapy 2 manual and participants materials into interviews to aid their recall. We asked them if there was anything they would change and so on. We also explored their views of the practicalities of implementing the intervention.
2	Methods: Thematic analysis Describe in more detail for readers (summarize in a few lines only).	We have added the following: "Interview transcripts were 'coded', where sections of text are assigned a descriptive label, producing dozens of codes per interview. These codes were then grouped into categories, and these were then grouped further into themes."
2	Did anything differ between sites of those that were interviewed compared to those that were not interviewed? Could their experience be different? Perhaps relate to size, location, etc. to contextualize for reader.	There were no apparent differences noted between the sites regarding the issues raised by trial participants and therapists. Participant recruitment was spread across the 17 urban and rural localities ranging from the north to the very south of England. All hospitals were established cancer care units with multidisciplinary teams (surgeons, oncology services, breast care nursing etc) although numbers of women treated annually varied from large hospitals in cities (Coventry, Wolverhampton, Milton Keynes) to those in more rural locations (Hereford, Taunton). Physiotherapy was not routinely provided for women undergoing non-reconstructive surgery. Therapists worked in outpatient departments and were not routinely involved in oncology services. We have added a sentence to the sample section to read: "There were no apparent differences between the sites regarding the issues raised by trial participants and therapists. We had a good range in terms of size and rural/urban sites across the 11 sites represented in the interview study."
22	Why did you decide to interview PT's in pairs? This point should be in the methods section and described in more detail. Could this have affected the responses of PT's?	We opted to interview therapists in pairs only where they worked together in a department. We trained at least two therapists in every recruiting centre. We opted to interview therapists in pairs to encourage discussion between the therapists and interviewer. We have added to the text: 'The physiotherapists who volunteered for the interviews worked closely together. Interviewing them in pairs allowed physiotherapists to share and reflect on their experiences, and aided recall where they had only treated a small number of participants, for example. It is possible that interviewing them in pairs could have affected their responses, but participants were remarkably candid about the challenges they experienced, thus we were not concerned that this was happening.'

2	Can you give more information on the PT's interviewed in chart as you did with the patient participants (age, years treating, number with previous oncology experience, etc. This will help link to your discussion of PTs previous experience with cancer patients and how this may have affected delivery of intervention.	Unfortunately we did not record detailed information on the physiotherapists such as age The sample section of the results includes all the information we have and reads: 'The physiotherapists had treated between one and 16 trial participants (median 5) and were based at hospitals that did not routinely provide postoperative physiotherapy after breast cancer surgery. They were experienced in the management of musculoskeletal conditions but did not currently work in breast cancer or oncology units. Some physiotherapists had experience of treating breast cancer patients presenting with problems such as restricted shoulder movement preventing the start of radiotherapy. One physiotherapist had past experience working on a cancer inpatient ward.'
2	Can you embed quotations within all parts the results section? I always find this brings more meaning and understanding to the results	We agree, and we have moved the quotes out of the tables and into the text. However, this means that the quotes are now included in the word count.
1 and 2	Also, link to participant (age, stage, etc.) and can analyze results based on some of these factors. Doing this once or twice within each section and then referring to table would improve this paper (like did in "Helping Myself" section) Try and summarize results more clearly based on intervention group; this was done well in some sections but was not clear in others.	We hope that the results section now more clearly identifies where subthemes are based on only intervention group, or both groups
2	Link results back to objective of acceptability; other aims clearly linked, but this aim should be more clearly addressed for readers. Also, how is acceptability defined, how do results link, etc.	In terms of how we defined acceptability, our approach was to understand the trial participants' experiences of the intervention, and to explore whether they found it helpful or enjoyable. We explored whether they were able to incorporate the intervention into their lives during cancer treatment. We hope that the results of the intervention arm demonstrate that the participants found the intervention to be useful and improved their wellbeing, although we recognise in theme 3 that some participants found it more difficult during chemotherapy. We have amended the first paragraph of the discussion to explicit mentioned acceptability.
2	Can touch on / explore more training of these physiotherapists. Described in good detail how they felt if working with this population was new. Now, what additional training is	Therapists attended a training day teaching them how to deliver the PROSPER programme. This included prescribing the exercises as well as behaviour change techniques to encourage adherence with the programme. Motivational Interviewing techniques were included along with case studies to demonstrate putting these skills into practice. A full description of

	needed? Did they have any MI training as well? If yes / no, why might this be important based on your results / discussion points?	the intervention has been published¹ in which these elements are described and we have included a brief description in the text. We have identified areas where physiotherapists felt additional training should be considered during future implementation which were related to breast cancer and its treatment and side-effects rather than skills such as MI. This is included in the final paragraph of the discussion. ¹Richmond H, Lait C, Srikesavan C, Williamson E, Moser J, Newman M et al. Development of an exercise intervention for the prevention of musculoskeletal shoulder problems after breast cancer treatment: the prevention of shoulder problems trial (UK PROSPER). BMC Health Serv Res. 2018;18(1):463. doi:10.1186/s12913-018-3280-
2	At times I feel like new ideas around results are presented in the discussion. Link points to themes more clearly.	We have now identified throughout the discussion where the points link to the subthemes

VERSION 2 – REVIEW

REVIEWER	Amy Dennett La Trobe University Australia
REVIEW RETURNED	14-Oct-2020

GENERAL COMMENTS	The authors have improved the quality of the manuscript. However some minor amendments need to be made prior to acceptance: Please use person first language throughout - please use term people with cancer or cancer survivors rather than cancer patients Abstract: Make sure objectives align with what is written in text - acceptability of the exercise intervention was not listed in the abstract Introduction Line 54 'loss of...' start new paragraph Results The revised version of the paper has too many quotes in text. These are repetitive of what is in the table. Please only include 1-2 quotes per theme at most. This will improve the flow of the manuscript Page 7 line 38-44 seems out of place and would be more appropriate in a setting section in the methods The way the themes are layed out make it difficult to work out what is a main and subtheme. I also notice that Delivering physiotherapy to reast cancer patients has an introductory sentence. Healing does not - keep the same pattern throughout Consider making the table supplementary material and adding a figure to represent the themes to make it clearer what the results are.
---

REVIEWER	Jenna Smith-Turchyn McMaster University, Canada
REVIEW RETURNED	21-Oct-2020

GENERAL COMMENTS	Thank you to the authors for updating this manuscript. They responded to my initial concerns in good detail. Additional comments (for editor and authors):  - It doesn't appear the SRQR checklist is completed (it is attached, but blank from what I can see) - Is the substantial increase in word count ok? This likely happened due to the increase of quotations, but I think the inclusion of these / the way this is formatted now is a huge improvement. If needed to cut back on words could paraphrase some of the quotes or take out a few in each section (so is still 1-2 in each section, but not 3-4) - Under "Data Collection" section, I am not sure what is meant by "the physiotherapists who volunteered for the interviews worked closely together"? Do you mean the ones who volunteered in pairs? Or did they all work closely together. Add a word or two to clarify this sentence. - Capitalize the start of all quotes (One off in "Making progress" section) - Updated table is an improvement and is easier to look at and relate to different stakeholders interviewed in this project.
--

VERSION 2 – AUTHOR RESPONSE

Review	Comment	Response
1	use person first language throughout	Thank you, we have edited the paper, particularly the methods section, to use first person language
1	use term people with cancer or cancer survivors rather than cancer patients	We choose not to use the term 'survivor' due to our own and others' research highlighting the problematic and alienating nature of this term. ¹ We have edited the paper so that we only use 'patient' in a few appropriate circumstances or in verbatim quotes. ¹ Rees, S. ' (2018) "What is a survivor? A qualitative exploration of the meaning of the term 'survivor' to young women living with a history of breast cancer." European Journal of Cancer Care 27:e12847 doi:10.1111/ecc.12847
1	Abstract: Make sure objectives align with what is written in text - acceptability of the exercise intervention was not listed in the abstract	We have added acceptability to the abstract
1	Introduction Line 54 'loss of...' start new paragraph	We have made this change
1	Results The revised version of the paper has too many quotes in text. These are	We have removed quotes and tried to ensure quotes in table are not repeated in the text where possible.

	repetitive of what is in the table. Please only include 1-2 quotes per theme at most. This will improve the flow of the manuscript	However, some quotes perfectly illustrate the theme in the table, but are also powerful when expanded on in the text. Nevertheless, we have restricted quotes wherever possible.
1	Page 7 line 38-44 seems out of place and would be more appropriate in a setting section in the methods	We have removed this extra information about physiotherapist training as it can be found in our paper describing the intervention (see Richmond et al., 2018)
1	The way the themes are layed out make it difficult to work out what is a main and subtheme. I also notice that Delivering physiotherapy to reast cancer patients has an introductory sentence. Healing does not - keep the same pattern throughout	We have changed the formatting of the headings to bold and italics and we hope that the formatting in the final article will make this clear. We have also added an introductory sentence to the theme 'Healing'. The second theme 'Being the perfect therapist' has an introductory sentence (but no subthemes).
1	Consider making the table supplementary material and adding a figure to represent the themes to make it clearer what the results are	We had firstly created a figure however this was complex and difficult to follow. We are keen to keep the table as feel that this is the best way of illustrating the results and providing all the detail. We seek advice from the Editorial team regarding whether this could be included within the main body of the text rather than as a Supplementary file.
2	It doesn't appear the SRQR checklist is completed (it is attached, but blank from what I can see)	We had completed the SRQR checklist but had not uploaded it with the revision, apologies. We have now updated and uploaded the checklist with this revision.
2	Is the substantial increase in word count ok? This likely happened due to the increase of quotations, but I think the inclusion of these / the way this is formatted now is a huge improvement. If needed to cut back on words could paraphrase some of the quotes or take out a few in each section (so is still 1-2 in each section, but not 3-4)	Thank you, we agree that this formatting is much better. We have removed/shortened some quotes
2	Under "Data Collection" section, I am not sure what is meant by "the physiotherapists who volunteered for the interviews worked closely together"? Do you mean the ones who volunteered in pairs? Or did they all work closely together. Add a word or two to clarify this sentence.	Thank you, we have clarified this so it now reads 'The physiotherapists who volunteered for the interviews and were interviewed in pairs worked closely together.'
2	Capitalize the start of all quotes (One off in "Making progress" section)	Thank you, we have checked and capitalised the quotes
2	Updated table is an improvement and is easier to look at and relate to different stakeholders interviewed in this project.	Thank you, we are pleased that the table works to illustrate the results. Thank you for all your comments which have helped improve the manuscript.